# Unlocking Flavor Potential Using Microbial β-Glucosidases in Food Processing

**DOI:** 10.3390/foods12244484

**Published:** 2023-12-14

**Authors:** Mariam Muradova, Alena Proskura, Francis Canon, Irina Aleksandrova, Mathieu Schwartz, Jean-Marie Heydel, Denis Baranenko, Liudmila Nadtochii, Fabrice Neiers

**Affiliations:** 1Molecular Mechanisms of Flavor Perception, Center for Taste and Feeding Behavior, INRAE, CNRS, University of Burgundy Franche-Comté, 21000 Dijon, France; alena.proskura@inrae.fr (A.P.); francis.canon@inrae.fr (F.C.); mathieu.schwartz@inrae.fr (M.S.); jean-marie.heydel@u-bourgogne.fr (J.-M.H.); 2International Research Center “Biotechnologies of the Third Millennium”, Faculty of Biotechnologies (BioTech), ITMO University, 191002 Saint-Petersburg, Russia; ivaleksandrova@itmo.ru (I.A.); l_tochka@itmo.ru (L.N.)

**Keywords:** β-glucosidases, flavor, glycoconjugates, aroma, taste, microbiota, food

## Abstract

Aroma is among of the most important criteria that indicate the quality of food and beverage products. Aroma compounds can be found as free molecules or glycosides. Notably, a significant portion of aroma precursors accumulates in numerous food products as nonvolatile and flavorless glycoconjugates, termed glycosidic aroma precursors. When subjected to enzymatic hydrolysis, these seemingly inert, nonvolatile glycosides undergo transformation into fragrant volatiles or volatiles that can generate odor-active compounds during food processing. In this context, microbial β-glucosidases play a pivotal role in enhancing or compromising the development of flavors during food and beverage processing. β-glucosidases derived from bacteria and yeast can be utilized to modulate the concentration of particular aroma and taste compounds, such as bitterness, which can be decreased through hydrolysis by glycosidases. Furthermore, oral microbiota can influence flavor perception by releasing volatile compounds that can enhance or alter the perception of food products. In this review, considering the glycosidic flavor precursors present in diverse food and beverage products, we underscore the significance of glycosidases with various origins. Subsequently, we delve into emerging insights regarding the release of aroma within the human oral cavity due to the activity of oral microbial glycosidases.

## 1. Introduction

Glycoside hydrolases (GHs) hydrolyze the glycosidic bond between two or more carbohydrates or between a carbohydrate and a noncarbohydrate moiety. These ubiquitous enzymes play a pivotal role in numerous physiological functions, such as the metabolism of cell-wall polysaccharides and glycolipids, glycan biosynthesis and remodulation, energy mobilization, defense mechanisms, and symbiotic interactions [1]. In the carbohydrate-active enzymes database (CAZY), GHs are classified into a growing number of families indicated by GH for glycoside hydrolases followed by a number for each specific family, such as GH1, GH2, GH3, GH5, GH9, GH30, and GH116 [2]. Multiple types of these enzymes are found due to the intricate nature of their glycoside substrates, which may comprise mono- or disaccharides. The terminal sugar of these glycosides can be β-d-glucopyranoside, α-l-rhamnopyranoside, α-l-arabinofuranoside, β-d-apiofuranoside, or β-d-xylopyranoside, while the central sugar in disaccharides is invariably β-d-glucopyranoside [3]. Enzymes that catalyze the cleavage of beta-linked polymers of glucose are called β-glucosidases (β-D-glucopyranoside glucohydrolases, E.C. 3.2.1.21). The GH1 and GH3 families include the most relevant β-glucosidase enzymes for biotechnological applications; GH1 contains the largest number of characterized β-glucosidases [2]. β-glucosidases can be classified into the following distinct classes based on their substrate specificity: aryl-β-D-glucosidases, which exhibit a high binding affinity for aryl-β-D-glucosides; cellobiases, which exclusively hydrolyze disaccharides; and broad-specificity glucosidases, which possess enzymatic activity toward a wide range of substrates and are the most frequently encountered β-glucosidases [4].

Microorganisms, particularly fungi and bacteria, are a large source of β-glucosidases [3]. Various species of *Aspergillus*, *Penicillium*, *Trichoderma*, and *Rhizopus* have been found to produce a multitude of β-glucosidases that have practical applications in the food and beverage industry to improve aroma [5]. In addition, glycosidases synthesize yeasts, such as *Saccharomyces cerevisiae*, which are utilized in the production of beer, bread, and wine [6]. Yeasts have been observed to produce β-glucosidase in various locations, including the cytosol, cell membrane, and cell wall [7]. However, the intracellular β-glucosidase activity of yeasts is not very valuable for industrial purposes. As a result, scientific investigations concerning yeast β-glucosidase activity predominantly target the enzyme’s presence in the whole cell and yeast supernatant, emphasizing the extracellular β-glucosidase activity [8].

The ability of β-glucosidases to release aromatic compounds from flavorless precursors is used in an increasing number of food products [9]. β-glucosidases are additionally widely utilized for the hydrolysis of flavanone glycoside, which is perceived as bitter. Naringin is a flavanone glycoside and one of the main compounds perceived as bitter in citrus juice. Naringin is hydrolyzed by β-glucosidases into naringenin and the disaccharide rutinose during juice extraction; as a result, bitterness can be reduced. The bitter threshold of naringin is lower than that of naringenin [10,11]. Within the flavor industry, β-glucosidases are the main enzymes involved in the enzymatic release of aromatic compounds from glucosidic precursors found in fruits and fermenting products, including wine, tea, and fruit juice [12,13].

Glycosides of volatile aromatic compounds that are water soluble and odorless are a useful source of flavor compounds [14]. Volatile aglycones can be released from glycoconjugates during fermentation via enzymatic catalysis as well as chemical methods, such as acid hydrolysis processes; thus, the aroma profile of foods is enhanced [9]. Some key aromas, such as terpenes, are unstable in acidic environments, limiting this approach to the release of terpene glycosides [9]. An alternative method employs rapid acid–heat hydrolysis, although undesirable aromas may be generated. While thermal hydrolysis can increase glycoside breakdown by up to 33%, elevated temperatures can also induce conformational alterations in the glycoside native structure [15]. In contrast, enzymatic hydrolysis specifically cleaves the glycosidic bond without altering the aglycone structure, making it a more convenient method [16]. Thus, efforts have been directed toward enhancing the flavor of juices and wines by hydrolyzing glycosidic aroma precursors with glycosidic enzymes, particularly β-glucosidase. The potential industrial applications of β-glucosidases have led to the exploration of new microbial enzyme sources that can be obtained with high structural stability in inexpensive fermentation media [17].

Food flavor results from the integration of information from various chemical senses, including olfaction (promoted by odorant molecules or aroma molecules when odorant molecules are perceived by the retronasal smell), gustation, and trigeminal [18]. The diversity of oral microbiota has been linked to individual differences in flavor perception [19,20]. Recent research has provided evidence that β-glucosidases originating from oral microbiota can enzymatically hydrolyze aroma precursors in various products, thereby releasing aroma molecules. These findings suggest that oral bacteria play a significant role in modulating aroma perception [21]. More than 750 bacterial species have been identified in the human mouth [22]. Among them, *Prevotella*, *Streptococcus*, *Veillonella*, *Neisseria*, and *Haemophilus* were proposed to produce the β-glucosidases involved in the release of aroma from food product during chewing [21,23,24]. The diversity of oral bacteria may influence the metabolization of aroma compounds and their precursors, leading to different retronasal olfactory responses [25]. Despite the importance of the oral microbiota in flavor perception, research on the connection between microbial communities, β-glucosidase activity, and the release of volatiles from flavor precursors is limited.

In this article, we will provide an overview of the applications of β-glucosidases in the food flavor industry and their microbial sources. We will also discuss the impact of these enzymes on human aroma perception. New knowledge on oral microbial enzymes could provide reasonable insights and help researchers design molecular tools that are optimized to produce aroma compounds or enhance the flavor intensity of specific food products. In addition, this knowledge could help food manufacturers optimize and modulate food processing.

## 2. Occurrence of Aroma Glycosides in Common Food

Glycosides are essential for the aroma of diverse plant-based food products, such as vanilla, fruit products, or beverages, such as tea, wine, and beer [26,27,28]. In many products, more aroma glycosides are present than their free aroma counterparts; for example, in many fruits, the level of glycosides was two to eight times greater than that of their free counterparts [29,30]. Given the low threshold and sensory characteristics of some aromas constituting aglycones, glycosidic compounds constitute a significant potential reservoir of flavor volatiles in fruit juice production [31].

In fruit products, the aroma precursors can be converted into volatile compounds during storage and processing, increasing the overall aroma of the resulting product [32,33]. Glycoconjugate-containing compounds are present in numerous fruits, such as grapes [34], apples [35], apricots [36], litchis [37], papayas [38], citrus fruits [39], pineapples [40], raspberries [41], kiwis [42], and peaches [43]. Depending on the type of glycoside, these compounds can generate citrus, green, floral, and sweet scents [44]. The fruit flavor is a result of a complex mixture of volatile compounds such as esters, ketones, terpenes, alcohols, and aldehydes [45]. Terpenoids, mainly C10 (monoterpenes) and C15 (sesquiterpenes), have been identified in the flavor profiles of many fruits [45]. Furthermore, most norisoprenoids, which are precursors of highly potent flavor compounds, are mainly present in fruit in glycosidic forms. In the green fruit of vanilla, fifteen aroma compounds in glucosyl form have been identified [46], and most of the vanillin is in glucosyl form [47]. The free vanilla aroma is increased during the development of the fruit and during the curing of vanilla beans [48]. In green tea, the hydrolysis of glycosides contributes to essential floral and sweet aromas of white teas, such as geraniol, linalool, and ionones [26,27,49,50,51]. Cis-3-hexenol is a widely recognized leafy alcohol in tea flavor, and this compound is found across all three tea varieties (black, oolong, and green teas); however, this compound is notably more abundant in green tea [52,53].

Ester compounds usually exhibit a sweet, floral, and fruity aroma. For instance, the nonalcoholic volatile glucoside methyl salicylate generates fresh and minty odors in green, oolong, and black teas [49,50]. Aldehydes can provide green, fresh, and citrus-like notes in foods [54]. A combination of these compounds was identified as important odor molecules that contribute to the distinctive rice flavor in rice koji [55]. The odor of 3-methylbutanal contributed to a pungent aroma in aged sake [56]. Additionally, the terpene 1,8-cineole could impart a fresh and mint-like aroma to red wines [57]. Infusion of spray-dried instant oolong tea led to a notable alteration in the tea’s scent profile via glycosides hydrolysis, resulting in an intensified expression of floral, fruity, and grassy notes. The primary odorants responsible for this effect were identified as furfural, cis-3-hexenol, geraniol, 2-methylbutanal, and 2-ethylfuran [58].

Glycosides significantly influence the sensory attributes of plant-based food products. This is primarily because a fraction of volatile compounds in these products are glycosidically bound, which is crucial for their application in the food processing sector [44]. In plants, aroma glycosides are generated through a process termed glycosylation. During this process, a sugar molecule (either an O-β-D-glucoside or a disaccharide, such as O-D-glycosides) binds to an aglycone compound that can generate an aroma [59]. This glycosidic linkage is established in a multistep enzymatic procedure. Initially, enzymes such as α-L-rhamnosidase, α-L-arabinosidase, or β-D-apiosidase hydrolyze the terminal sugar—such as rhamnose, arabinose, or apiose—resulting in the release of the corresponding β-D-glucosides. Following this, monoterpenol is released through the catalytic activity of β-D-glucosidase. Research on certain plant-based products, such as wine and fruit juices, has investigated the hydrolysis of monoterpenes due to their crucial role as flavor sources [60,61]. For instance, monoterpenes contribute to the floral and fruity aroma characteristics of wines [62] and cherry juice [14]. Furthermore, glycoside molecules in beverages, such as wine and beer, have been extensively studied.

Over 800 volatile and aromatic compounds have been identified in wine, encompassing those that originate from fruits (varietal aroma), those generated by yeast and bacteria during fermentation (enzymatic aroma), and those that develop during aging (post-enzymatic aroma). As a result, the flavor and aroma profile of a wine can vary widely due to many variations in winemaking [63]. Wine aroma is a multifaceted attribute influenced by numerous compounds that arise at different stages of winemaking. The aroma can be broadly categorized into the groups described below.

Primary aroma: this aroma is determined by the grape variety and is influenced by aromatic glycosides present in the grape skins as nonvolatile and odorless glycoconjugates [63]. Several authors highlight the prominent role of terpenes in the varietal aroma due to their transformative capabilities [64].

Fermentation or secondary aroma: this fermentative aroma arises from various precursors during alcoholic and malolactic fermentation. During these processes, glycosides undergo enzymatic hydrolysis, releasing volatile aroma compounds that contribute to the distinct flavors of wine. Notably, wines contain flavor precursors such as linear or cyclic alcohols, including hexanol, phenylethanol, and benzyl alcohol [65].

Bouquet or tertiary aroma: this post-fermentative aroma originates from enzymatic and physicochemical processes, including oxidation and reduction, throughout the conservation and aging phases of wine [64].

The complex and unique flavor profiles of wines are due to aromatic glycosides that occur as nonvolatile and odorless glycoconjugates in the skin of grapes [65]. Considering the glycoside content, most aromatic aglycones are associated with apiosyl glycosides, which are prevalent in grape juice, constituting up to 50% of the grape variety. Following this, rutinosides range from 6% to 13% and glucosides from 4% to 9%. The glycoside concentrations and their variations are largely contingent upon the specific grape variety. Glycosides in wine originate from grapes during ripening and may correlate with the concentration of their respective aglycones [66]. During the fermentation and aging processes, these glycosides are enzymatically hydrolyzed, releasing volatile aroma compounds that contribute to the wine’s characteristic flavors. In a separate but related process, terpenes play a significant role in the varietal aroma of wines due to their ability to transform into other compounds [67]. These terpenes, which may influence the flavor of wine, can generate other flavor precursors, such as linear or cyclic alcohols (hexanol, phenylethanol, benzyl alcohol) [45].

*Humulus lupulus* of the family *Cannabaceae* is a major raw material for beer. Among the aroma compounds responsible for the aroma of beer, those derived from *Humulus lupulus* include monoterpenes typified by linalool. On the cone surface of *Humulus lupulus*, there are trichomes called lupulin, which are specifically differentiated to accumulate aroma components therein. Many aroma compounds accumulate within lupulin [68]. The female cones of hops are used to impart flavor, aroma, antimicrobial properties, and foam stability to beer [69]. Studies of hop aroma have mostly focused on volatile compounds in the essential oil. While these volatiles are the main contributors to hop aroma in beer, glycosidically bound aroma precursor compounds also play a role [70]. Hops contain several classes of aglycone volatile molecules, including monoterpene alcohols, monoterpene polyols, norisoprenoids, aliphatic alcohols, and volatile phenols. Terpenes and monoterpene alcohols are found in hops in free form as well as bound to other molecules [71]. Extensive research on the glycosidic composition of hop cones has focused mainly on monoterpene alcohols, compounds often considered to generate aroma in some hop varieties, and norisoprenoids such as β-damascenone [72]. Hop aroma glycosides play a role in the aroma development of beer [73]. In brewing, hops (*Humulus lupulus*) are an important component and contain α-acids, which prevent spoilage and give the beer a special characteristic flavor [74]. Linalool, 3-methylbutanoic acid, myrcene, and dimethyl trisulfide are the most influential aromatic compounds, and other glycosides, such as geraniol, also play an important role [75]. These aroma glycosides are odorless but release volatile aromatic compounds when hydrolyzed by high temperature, low pH, and yeast enzymes [76]. The compounds promote aroma development at various stages of brewing and beer storage [77].

### 2.1. Yeast β-Glucosidases

Yeast, as one of the primary microorganisms responsible for the production of glucosidase enzymes in the food industry, plays a pivotal role [78]. Yeasts can be used during the fermentation of fruit juices, wine, and other plant-based products [79]. Yeast contributes to the aroma development of many products, such as chocolate [80] or sake [81], which require fermentation processes, through glycosidase activities. However, the yeast glycosidases involved in aroma development have been mainly studied for wine and beer.

The yeasts that are present during spontaneous fermentation can be classified into the following categories: *Saccharomyces* yeasts, primarily *Saccharomyces cerevisiae*, and non-*Saccharomyces* yeasts, which encompass members of the genera *Rhodotorula*, *Pichia*, *Candida*, *Debaryomyces*, *Metschtnikowia*, *Hansenula*, and *Hanseniaspora* [82]. *Saccharomyces cerevisiae* is widely used in the conversion of grape aroma precursors to varietal wine aromas [83]. β-Glucosidase production has also been observed in yeasts belonging to the genera *Candida* [84], *Debaryomyces* [85], *Hanseniaspora*/*Kloeckera* [86], *Kluyveromyces* [87], *Metschnikowia* [88], *Pichia* [89], *Saccharomycodes*, *Schizosaccharomyces*, and *Zygosaccharomyces* [90]. For instance, Kai Hu et al., showed that isolates of *Hanseniaspora uvarum*, *Pichia membranifaciens*, and *Rhodotorula mucilaginosa* among 493 non-*Saccharomyces* isolates exhibited high β-glycosidase activity. The *Hanseniaspora uvarum* strain was shown to exhibit catalytic specificity for aromatic C13-norisoprenoid glycosides and some terpenes, enhancing fresh floral, sweet, berry, and nutty aroma characteristics in wine production [91]. Non-*Saccharomyces* species, such as *Candida lusitaniae*, *Hanseniaspora guilliermondii*, *Metschnikowia pulcherrima*, and *Pichia anomala*, exhibited the highest β-glucosidase activity to improve wine aroma complexity [92]. β-glucosidases from strains *Hanseniaspora* sp. and *Pichia anomala* were used to release glucoside-linked monoterpenes, which are the major contributors to floral and fruity aromas in Muscat-type wines, at the final stage of alcoholic fermentation [93]. Additionally, β-glucosidase from *Sporidiobolus pararoseus* was purified and characterized with potential application for terpene compound release in wine [94]. β-Glucosidase from the yeast *Brettanomyces anomalus* hydrolyzed glycosides from cherry beers to produce more benzyl alcohol and eugenol, resulting in a typical cherry beer aroma. The enzyme also hydrolyzed forest fruit milk glycosides to release more methyl salicylate, which imparted a more desirable spicy flavor to the beverage compared with that of commercial almond β-glucosidase [95].

Furthermore, non-*Saccharomyces* species are strategically used to create multistarter, mixed, or sequential cultures in combination with *Saccharomyces cerevisiae* [96,97,98]. This combination can improve the flavor complexity and characteristics of wines because non-*Saccharomyces* yeasts are rich in various enzymes that can hydrolyze and release abundant aroma compounds [99,100]. For example, Wen-Ke Shi et al. found that cofermentation with *Pichia kudriavzevii* and *Saccharomyces cerevisiae* enhanced wine flavor and quality. This enhancement was attributed to a reduction in volatile acidity and an increase in aroma compound content when compared with fermentation using only *Saccharomyces cerevisiae* [101]. Recent studies have verified that the elevated levels of terpenes in wines resulting from mixed cultures of *Saccharomyces cerevisiae*/*Hanseniaspora guilliermondii* with sequential fermentation could considerably boost the polyphenol and volatile aroma component contents in Nanfeng tangerine wines [102]. Additionally, cocultivation of the β-D-glucosidase-producing strain *Debaryomyces pseudopolymorphus* with *Saccharomyces cerevisiae* VIN13 during Chardonnay juice fermentation was observed to significantly elevate the concentrations of citronellol, nerol, and geraniol [103]. Equally, mixed cultures of Sauvignon Blanc grape juice with *Candida zemplinina*/*Saccharomyces cerevisiae* and *Torulaspora delbrueckii*/*Saccharomyces cerevisiae* generated wines with high concentrations of terpenols compared with wines fermented with *Saccharomyces cerevisiae* [104]. However, some studies have shown that non-*Saccharomyces* yeasts can be inhibited by *Saccharomyces cerevisiae* or the vinification environment [105,106,107].

The complexity of detecting enzyme activity is further demonstrated by the in vitro substrates used for β-glucosidase detection in yeasts. These substrates undergo hydrolysis by glucanases; thus, the potential overlap of these two enzymatic activities must be considered [108]. *Saccharomyces* is known to exhibit diverse exo-1,3-β-glucanases encoded by genes such as EXG1 and its paralogs SPR1 and EXG2. The catalytic activities of these enzymes have been associated with glycoside hydrolysis [109]. According to the *Saccharomyces* Genome Database, the EGH1 gene encodes a β-glucosidase with a broad specificity for aglycones. This enzyme has shown the capability of hydrolyzing flavonoid glucosides such as 7-O-β-glucosides or 4′-O-β-glucosides of flavanones, such as NAR 7-gluc, flavones, flavonols, and isoflavones [110]. This activity could contribute to aroma enhancements.

### 2.2. Bacterial β-Glucosidases

Lactic acid bacteria (LAB) represent a noteworthy group of microorganisms that play an important role in food production [111]. LAB grow in diverse nutrient-rich environments, such as animal (milk, meat) or plant-based products (fruits, cereals). Incubation with lactic acid bacterial strains increases the nutritional value and improves the organoleptic qualities of these products, such as fruit juices such as blueberry, citrus, apple, and elderberry [112,113,114,115]. Microbial strains possess the ability to metabolize sugar for the production of lactic acid and display diverse catalytic activities, leading to the generation of distinct aroma-active compounds derived from their precursors [116]. The main aroma compounds released by bacteria include aldehydes, organic acids, higher alcohols, esters, carboxylic acids, and ketones [117,118]. Multiple bacterial species, such as *Leuconostoc*, *Lactobacillus*, and *Pediococcus*, exhibit glucosidase activity and are commonly used in the food industry to improve flavors and aromas in fermented foods and beverages (Table 1). The *Lactobacillus* genus is widely used in the fermentation of various food products, including cheese [119], yogurt [120], and pickles [121], since it possesses a diverse range of enzymes, such as alpha-amylase, beta-glucosidase, and beta-galactosidase. For example, *Lactobacillus pentosus* can metabolize complex carbohydrates, such as starch, cellulose, galactan, xylan, pullulan, pectin, and gums [122]. *Lactobacillus harbinensis* and *Pediococcus pentosaceus* with a high content of β-glucosidase have been applied to improve the antioxidant properties, flavor profile, and quality of functional bread; the high content of LAB-producing β-glucosidase increases the functionality of the kiwi fruit substrate [123].

There is considerable interest in the β-glucosidase activities of LAB that conduct the malolactic fermentation of wine. Secondary fermentation by *Lactobacillus* can modify the aromatic profile of wines by releasing significant concentrations of diacetyl-(2,3-butanedione) and other carbonyl compounds derived from citric acid, which contribute to the oiliness and aroma of wines [124]. However, since high ethanol content often causes inhibition and slows the process of MLF in wine, the selection of highly ethanol-tolerant strains is a significant goal for wine producers. For example, Xiaonan Li et al. produced a β-glucosidase, Lb0241, derived from *Lactobacillus brevis* T61, which exhibits strong resistance to acids and ethanol and inhibited β-glucosidase activity. This β-glucosidase was shown to effectively hydrolyze aromatic precursors and increase the level of aromatic compounds in strawberry wine [125]. β-Glucosidase with high hydrolytic efficiency for terpenyl glycoside has been reported from *Sporidiobolus pararoseus* [94] and *Aureobasidium pullulans* [126], suggesting their potential application for the development of wine aroma. The genus *Oenococcus* is also widely used in the malolactic fermentation of wines. The *Oenococcus oeni* species is generally beneficial for the final aroma compounds in wine. This bacterium is involved in malolactic fermentation (MLF) due to its resistance to ethanol and low pH. The selection of strains of *Oenococcus oeni* that exhibit glycosidase activity may promote flavor enhancements in winemaking [127]. Strains *Oenococcus oeni* MS20 and MS46 were revealed to show higher activity of β-glucosidase to degrade synthetic glycoside (eriodictyol 7-O-β-rutinoside), while low alpha-L-rhamnose activity was induced, representing better performance of MS46 in a wine-like medium containing 6% ethanol at pH 4 [128].

**Table 1 foods-12-04484-t001:** Applications of bacterial β-glucosidase in foods.

Food	Microorganism	Effect	Reference
Soymilk fermented with kombucha and fructo-oligosaccharides	*Lactobacillus rhamnosus*	The increased β-glucosidase activity promoted favorable flavor substances, such as citric acid and linalool.	[129]
Cashew apple juice (CAJ)	*Lactobacillus plantarum*, *Lactobacillus casei*, *Lactobacillus acidophilus*	During CAJ fermentation by lactic Bacteria, the fruity odor decreased, while whiskey and acid odors were elevated. This activity is referring to β-glucosidase action derived from lactic acid species.	[130]
Peanut milk	*Lactobacillus delbrueckii* ssp. *bulgaricus* and *Streptococcus* *salivarius* ssp. *thermophilus*	A significant decrease in green/beany flavor and a significant increase in creamy flavor occurred due to the action of β-glucosidase derived from lactic acid bacteria.	[131]
Chickpea milk	*Lactobacillus plantarum*	The hydrolysis of glucosides of soy-based products using β-glucosidase increased their bioavailability, resulting in changed aroma profiles that impart fruity and creamy characteristics	[132]
Soymilk	*Lactobacillus plantarum* 70810	In comparison with original soymilk base, the concentrations of the characteristic flavor compounds for fermented soymilk using *L. plantarum* 70810 increased, whereas hexanal, 2-pentylfuran, and 2-pentanone in relation to the beany flavor of soymilk decreased.	[133]
Sicilian table olives	*Lactobacillus paracasei*, *Lactobacillus plantarum*	These reduced the debittering time during the fermentation of Sicilian table olives and caused an increase in hydoxytyrosol, tyrosol, and verbascoside compounds.	[134]
Tannat wine	*Oenococcus oeni*	Under the action of β-glucosidase, a significant increase in 2-phenylethanol and the sum of terpenols (linalool, R-terpineol, nerol, and geraniol) brought a flavor richness to wine.	[135]
Peeled frozen tomatoes	*Lactobacillus plantarum*	Several terpenes important for aroma profiles released included (Z)-geraniol, 6,7-dihydrogeraniol, melonol, linalool, and D-limonene after β-glucosidase treatment.	[136]

Some strains of *Leuconostoc* bacteria are known to produce diacetyl, which generates buttery and creamy flavors, and 2,3-butanedione, which gives a nutty and caramel-like flavor. Yu Geon Lee et al., found that *Leuconostoc mesenteroides* enhanced flavonoid aglycone content during the fermentation of onion via β-glucosidase action [137]. Marta Acin-Albiac et al. produced and characterized 6-phospho-β-glucosidases from *Leuconostoc pesudomesenteroides* and *Lactobacillus plantarum* during fermentation of brewers’ spent grain (BSG). Researchers observed increased metabolic activity of conjugates of gentiobiose, cellobiose, and β-glucoside with phenolic compounds, achieving efficient reduction/modification of cellulose derivatives and phenolic compounds in BSG [138].

Additionally, other non-lactic species, such as *Bacillus* spp., demonstrated β-glucosidase activity to hydrolyze β-glucosidic linkages between carbohydrate residues in aryl-amino- or alkyl-β-D-glucosides, cyanogenic glucosides, short-chain oligosaccharides, and disaccharides [139]. For instance, *Bacillus amyloliquefaciens* and *Bacillus licheniformis* released monoterpenoids and benzoids, of which the most released compounds were neric acid and 2-phenylethanol in grapes, respectively [140]. Other research on *Bacillus subtilis* contributed to knowledge on the complex factors that influence the unique flavor and nutritional value of natto by hydrolyzing isoflavone glycosides [141].

### 2.3. Other Fungal β-Glucosidases

Mushrooms, which are mostly saprophytic fungi, derive their nutrition from the adjacent cellulosic material and hence contain lignocellulolytic enzymes, such as cellulases, β-glucosidases, lignin peroxidases, and laccases [142], in addition to a high number of detoxification enzymes, such as glutathione transferases [143,144,145]. β-glucosidases are primarily present in the cellulase enzyme system in fungi and are responsible for hydrolyzing short-chain oligosaccharides and cellobiose into glucose. This hydrolysis process is a rate-limiting step facilitated by the synergistic action of endoglucanases and cellobiohydrolases [146].

The application of fungal enzymes in food production due to the presence of β-glucosidase offers a promising approach to enhance the organoleptic and nutritional content of foods [147]. Furthermore, glycosidases from fungi are more resistant to heat than those from plant and yeast origin. The activity of fungal β-glucosidase is subject to inhibition by glucose; however, the exoglycosidase activities of the enzyme remain relatively stable when exposed to the concentrations of glucose found in fruit juices [148,149]. Some fungal species exhibit strong activity even at low pH, making them particularly useful for applications in plant-based products and wine processing. The ability of these enzymes to work efficiently under acidic conditions is advantageous for maintaining the flavor and aroma of different foods [150]. In the context of flavor enhancement, β-glucosidase enzymes have been produced, purified, and characterized across numerous fungal species. The preeminent contributors to this body of research encompass a wide taxonomic range, including, but not limited to, representatives of *Aspergillus* spp. [11,151], *Candida molischiana* [152], *Trichosporon asahii* [153], *Rhizopus stolonifer* [154], *Mucor plumbeus* [155], *Aureobasidium pullulans* [126], *Trichoderma* [156], and *Penicillium italicum* [157]. Some examples of food applications of β-glucosidase are presented in Table 2.

**Table 2 foods-12-04484-t002:** Applications of fungal β-glucosidase in foods.

Food	Microorganism	Effect	Reference
Passion fruit juice	*Aspergillus niger*	Linalool, benzyl alcohol, and benzaldehyde levels increased in a passion fruit juice.	[158]
Fruit juices and wines	*Candida molischiana*	Increased the levels of linalool, benzyl alcohol, and 2-phenylethanol in wine and some fruit juices under β-glucosidase action.	[152]
Red grape Cabernet Gernischt (*Vitis vinifera* L. cv.)	*Trichosporon asahii*	Volatile flavor compounds in the β-glucosidase-treated samples were significantly increased.	[159]
Strawberry glycosidic extract	*Aspergillus aculeatus*, *Aspergillus foetidus*, *Aspergillus japonicus*, *Aspergillus niger*, *Aspergillus tubingensis*	Liberated volatiles (2,5-dimethyl-4-hydroxy-3(2H)-furanone (furaneol)) or cinnamic acid contributed to the flavor of fresh strawberries.	[160]
Fermented glutinous rice	*Aspergillus cristatum*, *Aspergillus niger*, *Aspergillus oryzae*	Modified the flavor metabolism in rice by releasing linalool oxides, β-ionone, and geraniol compounds.	[161]
Fu brick tea	*Aspergillus*, *Candida*, *Debaryomyces*, *Penicillium*, Unclassified_k_Fungi, Unclassified_o_ *Saccharomycetales*	Linalool, acetophenone, and methyl salicylate were identified as key volatiles contributors to the “fungal flower”, “flower”, and “mint” attributes; cedrol contributed to the “woody” attribute, and twelve alcohols and aldehydes were related to the “green” attribute during the manufacturing process.	[162]
Green tea	*Aspergillus niger*	Increased concentrations of cis-3-hexenol, hexanol, geraniol, and benzyl alcohol.	[163]
Craft beer	*Candida glabrata*	Glucoside-binding terpenes provided the final beer product with unique floral and fruity characteristics.	[164]
Soy sauce	*Aspergillus oryzae*	Formation of flavors such as alcohols, acids, esters, aldehydes, furans, and pyrazines during soy sauce fermentation.	[165]

Modulating coffee aroma via fermentation by *Rhizopus oligosporus* led to positive impacts on coffee aroma, imparting it with desirable fruity notes [166]. Fermentation with *Rhizopus oligosporus* has been determined to improve the flavor characteristics of sweet potato. In particular, the released alcohols, aldehydes, and esters increased the characteristic fatty odor, floral, fruit, and sweet flavors of sweet potato residue [167]. High levels of 1-hexanol, hexanal, 3-octanone, and methyl linoleate were also observed in the moromi fermentation product, with *Aspergillus sojae* synthesizing soy-sauce-specific flavors [168]. *Candida* and *Issatchenkia* fungal species were shown to release β-glucosidase to elevate terpenes, esters, and fatty acids and thus enhance flavor complexity in wines [84,169]. Another recent study showed that *Issatchenkia* and *Candida* were positively correlated with 1-hexanoic acid, caprylic acid, 2-methoxy-4-methylphenol, 5-pentyldihydro-2(3H)-furanone, 1-nonanoic acid, acetic acid, (2Z)-2-phenyl-2-butenal, and 4-sec-butylphenol in *Hongqu* aromatic vinegar [170]. This is probably due to the β-glucosidase activity exhibited by these fungal species.

It has been observed that solid-state fermentation with filamentous fungi can enhance the total polyphenol content in matrices of beans, grains, and fruits, which is attributed to the production of enzymes that facilitate the release of phenolic compounds bound to the cell wall [171]. For instance, solid-state fermentation (SSF) by *Aspergillus oryzae* and *Mucor plumbeus* in green canephora coffee could produce caffeylquinic acid (CQA) and polyphenols in fermented beans for the modulation and improvement of coffee flavor [155].

### 2.4. Plant β-Glucosidases

β-Glucosidases are ubiquitous among plants and are responsible for the hydrolysis of complex carbohydrates, including starch and cellulose, into simple sugars. Plants contain β-glucosidases belonging to GH1, GH3, and GH5 [58,172,173], of which GH1 is the most characterized [2]. These enzymes play a crucial role in several biological processes in plants, including growth, development, and response to environmental stress [174]. During endosperm germination, β-glucosidases and other glycosidases and glycanases degrade the plant cell wall, leading to the formation of intermediates for cell-wall lignification. Additionally, β-glucosidases are involved in plant defense mechanisms by catalyzing the hydrolysis of glycosides and releasing several chemical defense compounds, including phytohormones, flavonoids, and cyanogenic glucosides [2].

β-Glucosidase is common in plant-based products, such as grape [175], almond [176], maize [177], melon [178], orange [179], papaya [180], sweet cherry [181], wheat [182], and rice [183]. The enzymatic activities of endogenous β-glucosidase tend to increase progressively as fruits ripen [32]. Plant β-glucosidases play an important role in diverse biochemical transformations that occur during food processing. Specifically, endogenous β-glucosidases act as catalysts in the conversion of aroma precursors present in vanilla beans, leading to the formation of aromatic aglycones throughout the treatment process [184,185]. Similarly, in the context of tea processing, endogenous β-glucosidases participate in the development of the characteristic floral aroma by facilitating the hydrolysis of glycosides of monoterpenes and aryl alcohols, thereby yielding the corresponding aglycones. These enzymatic activities substantially contribute to the overall flavor profile of the final food products and underscore the significant role of plant β-glucosidases in food science and technology [186]. In grapes, the most abundant glucosidase activity is that of β-glucosidase, which is present alongside other endogenous grape glycosidases, such as α-arabinosidase, α-rhamnosidase, and possibly β-apiosidase [60]. Ziya Günata and coauthors carried out a comprehensive and pioneering study into the pivotal roles played by apiosidases and acuminosidases (either sequentially or in cooperation with exogenously introduced and endogenous β-glucosidases) in the intricate process of aroma formation during wine production [32,148,187].

Plant glucosidases typically exhibit their optimal pH activities within the range of 4.0 to 6.0. However, in fruit juices with lower pH levels, the observed activity of most glycosidases is approximately 5% to 15% of their maximum potential [175]. Nevertheless, the activity of glucosidase produced by most plants can be inhibited in low pH conditions, with glucose, and by thermal treatment [188,189]. However, a glucose-tolerant β-glucosidase has recently been characterized in black plum and *Prunus domestica* seeds. This enzyme exhibited its highest activity at a temperature of 55 °C and a pH of 5.0 [190]. Almond β-glucosidase has been well investigated to evaluate the potential application of the enzyme as a biocatalyst under high pressure conditions. Scientists have shown that almond β-glucosidase is relatively more thermostable than other β-glucosidases derived from vanilla [184], grape [175], maize [177], and tea leaf [186]. Furthermore, in comparison to β-glucosidases derived from microbial sources, β-glucosidases of plant origin show distinctive attributes, such as exceptional glucose tolerance, a wide range of substrate specificity, and remarkable selectivity [191]. As a result, an array of β-glucopyranosides has been successfully synthesized by employing β-glucosidases sourced from plants [192,193].

### 2.5. Oral β-Glucosidases in Glycoside Metabolization

Scientific investigations have substantiated the presence of β-glucosidase within the oral cavity, indicating the potential hydrolysis of aroma glycosides to modulate flavor perception by the liberation of glycoconjugates. In humans, several β-glucosidases were identified, as well as some of their physiological substrates. Additionally, a β-glucosidase termed cytosolic β-glucosidase was shown to catalyze the hydrolysis of a broad range of dietary xenobiotic glycosides [194]. However, this human enzyme is present in the liver, kidney, intestine, and spleen and was not found in human saliva. Moreover, recent publications highlight the significant involvement of the human oral microbiota glycosidase in the modulation of flavor perception and sensitivity [195,196,197]. Odorless fruit glucosides have been shown to release active odorant molecules when incubated with oral microorganisms (terpenes, benzenic compounds, and lipid derivatives) [23].

Different bacterial species can produce β-glucosidase to metabolize precursor compounds present in foods to generate aroma molecules in the mouth. The most abundant bacterial taxa in the oral cavity are classified into the phyla *Firmicutes*, *Proteobacteria*, *Actinobacteria*, *Bacteroidetes*, and *Fusobacteria* [23]. However, the diversity of bacteria varies across different oral cavity regions; saliva and tongue microbiota exhibit the greatest diversity and are dominated by genera such as *Rothia*, *Prevotella*, *Streptococcus*, *Veillonella*, *Fusobacterium*, *Neisseria*, and *Haemophilus* [198,199]. In recent years, several studies have provided evidence for a strong link between flavor perception and human oral microbiota [200]. Different enzymatic activities promoted by these oral bacteria, including glycoside hydrolysis, contribute to the modulation of food product flavors [201].

Glycoside conjugates can be metabolized by bacteria such as *Prevotella* and *Veillonella* species, which are consequently associated with increased glycoside hydrolysis [25]. This reaction leads to the release of aroma compounds, such as terpenes, aromatic derivatives, or alcohols [202,203]. *Actinomyces naeslundii* produced linalool and its associated oxides, indicating that this microorganism may play a significant role in the biosynthesis of floral aromas derived from grape glycosides [23]. It has been shown that eugenyl-β-D-glucoside can be hydrolyzed by oral microorganisms, resulting in the release of the free form of eugenol-aglycone. The authors propose that their results may indicate that not only eugenyl β-D-glucoside but also all glycosides with a β-D bond can undergo a similar hydrolysis process upon incubation with oral microorganisms [204]. A recent study on the microbiome of the tongue dorsum has revealed that a greater prevalence of *Actinomyces* bacteria is positively associated with heightened taste sensitivity in wine tasters. Additionally, a negative correlation has been observed between the abundance of the genus *Gemella* and career length in wine tasters, indicating a potential inhibitory effect on wine taste perception [205].

Saliva activity can lead to the release of monoglucosides and disaccharide glycosides, which is very dependent on the glycoside type and the individual. For example, it was shown that flavonoid glycosides were hydrolyzed when incubated with saliva within minutes [206]. The hydrolysis process is also greatly influenced by a low pH and high alcohol or glucose concentrations [203]. Despite the buffering and alcohol-diluting properties of saliva, coupled with the rapid temperature elevation post-ingestion, glycoside hydrolysis may continue as long as residual wine remains within the oral cavity. The relatively brief residence time of wine in the oral cavity suggests that oral microbiota have a limited impact on the perception of wine aroma. However, recent research findings suggest that nonvolatile wine matrix compounds and oral and pharyngeal mucosa may interact, which may increase the residence time of aroma precursors and free aroma compounds in the oral and pharyngeal cavities; as a result, their susceptibility to oral factors, such as saliva and oral microbiota, is increased [207]. The continuous replenishment of wine during in vivo consumption suggests that oral microbiota might indeed modulate wine aroma perception [208]. Moreover, the composition of the wine matrix significantly influences the intraoral aroma release of specific odorants under physiological conditions [209].

Another study of guaiacyl glucoside, geranyl glucoside, and glycosides extracted from Gewürztraminer grapes also revealed significant interindividual variation in glycoside flavor perception [210]. The considerable individual variability in response to these glycosides confirms earlier reports that a key factor contributes to differences in sensory perception among individuals. This variation was attributed by the authors to oral microbiota, which produce the majority of glycosidase enzymes in the mouth [211], as well as variations in the prevalence and degree of activity of mouth β-glucosidase [212]. Olfactory detection thresholds are the principal factor that governs flavor perception from these glucosides [25].

Due to the potential association between microbiota composition and an individual’s changes in olfactory and gustatory perception, exploring the relationship between microbiota and chemoperception is a significant challenge. Therefore, further studies are needed to elucidate these complex interrelationships. Additionally, identification of the enzymes responsible for the release of aromatic aglycones by microbial enzyme hydrolysis will be valuable.

## 3. Trends and Prospects

β-Glucosidases typically originate from endogenous sources within plant tissues and function as catalysts for the hydrolysis of flavor precursors, liberating the aglycone moiety. Glycosidase enzymes can hydrolyze the glycosidic linkage to release the aglycone. This enzymatic process provides opportunities for the incorporation of exogenous enzymes into food and beverage production processes at various stages, with the aim of enhancing aroma and flavor characteristics.

In the flavor industry, β-glucosidases play an indispensable role as enzymatic facilitators in the hydrolysis of glucosidic precursors, prominently encountered in fruits and fermented products, thereby culminating in the liberation of aromatic compounds. This enzymatic cascade holds substantial potential for applications within the food processing sector, particularly as a pivotal flavor-enhancing enzyme in products such as wine, tea, and fruit juice.

The β-glucosidase enzyme has emerged as a linchpin for flavor enhancement, with a multitude of β-glucosidic flavor precursors being widespread across diverse biological domains, including plants, bacteria, fungi, and yeasts. The hydrolytic action of β-glucosidases on these compounds is paramount in elevating the quality of foods and beverages derived from these sources. This process entails the enzymatic cleavage of glucoside bonds and the liberation of aglycones, which are aromatic precursors that are intricately bound to sugar molecules; as a result, the compounds are accessible for flavoring foods and beverages. Given the extensive diversity of glycosidases, a significant challenge lies in deciphering the specificities related to the various potential glycoside substrates. Identifying the respective genes and corresponding enzymes is crucial in fermentative and oral microorganisms. This endeavor not only expands our comprehension of glycosidase functionality but also establishes a foundation for further investigations into the complex interactions and roles these enzymes perform within microbial communities.

Although food and beverage manufacturers extensively employ glucosidases to enhance flavor during production processes, the comprehensive extent of these enzymes’ presence and their potential functions within the human oral cavity continue to be subjects of investigation. Glycosidases have emerged as promising subjects of study to decipher the intricate mechanisms that govern flavor perception within the oral cavity. Regrettably, research on oral microbiota enzymes within individual subjects and their direct impact on flavor perception has been limited.

Based on the knowledge acquired concerning oral microbial glucosidases, these compounds show substantial potential, particularly for the design of tailored molecular tools to improve the flavor intensity of specific food products or specifically augment the production of aroma compounds. Further investigation is imperative to elucidate the distinctions between the roles played by saliva and tongue microbiota in shaping flavor perception. It is reasonable to posit that metabolites situated in close proximity to taste buds may exert a more pronounced influence on taste perception.

Future studies should aim to characterize the glycosidases of the oral microbiota, especially those produced in sufficient quantities to act rapidly on the glycoside substrates present in foods. Each of these glycosidases is likely active on a specific set of substrates, different from others that remain to be characterized. These analyses will certainly help better anticipate, on one side, the aroma release in the mouth based on the nature and quantity of aromatic glycosides present in the food, and on the other side, the specific microbiota of each individual.

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
