# Peer review of "Unlocking Flavor Potential Using Microbial β-Glucosidases in Food Processing"

_foods, 2023, doi:10.3390/foods12244484_

Round 1
Reviewer 1 Report
Comments and Suggestions for Authors
Thank you for the opportunity.
The work is good but rearrangement and some few modifications are required.

This is an interesting review. I acknowledge the modern English used.
Some suggestions are within the manuscript that may improve the work.
Author Response
We want to thank Reviewer 1 for his great job in improving the paper. All the proposed modifications in the PDF file have been taken into account.

Reviewer 2 Report
Comments and Suggestions for Authors
Dear Authors,
The review titled "Unlocking flavor potential: The role of microbial β-glucosidases in enhancing flavor perception during food processing and in the mouth" is, in my opinion, complete, therefore I rated this manuscript positively. In general, the review is good to read, the structure of the work is clear.
1. The subject matter of the review is consistent with the profile of the Foods.
2. The manuscript is written in understandable language for reader.
3. In the Review the Authors analyze the role of glycosides as aroma precursors. The Authors analyze also insights regarding the release of aroma within the human oral cavity due to the activity of oral microbial glycosidases.
4. After reading the paper, it is clear that the authors analyzed over two hundred literature items on the topic.
This review, a collection of existing information from the literature, provides readers with the most important information on the topic in a nutshell.
5. There are two tables in the Review, which are a collection of information related to the topic of the Review. The tables contain important information that is supported by literature references.
However, to improve the quality of the work, I suggest that the authors consider adding to the Review several of the latest works by other researchers that were published this year. I hope that the suggested literature will be useful and will improve the substantive quality of the Review.
For example:
Volatilomics Analysis of Jasmine Tea during Multiple Rounds of Scenting Processes.
https://doi.org/10.3390/foods12040812
Derivation of Kokumi γ-Glutamyl Peptides and Volatile Aroma Compounds from Fermented Cereal Processing By-Products for Reducing Bitterness of Plant-Based Ingredients. https://doi.org/10.3390/foods12234297
Author Response
Thanks for your positive comments. We have added the two proposed references.

Reviewer 3 Report
Comments and Suggestions for Authors
In the present manuscript, the Authors summarize the existing knowledge about the role of microbial β-glucosidases in enhancing the food aroma through the cleavage of non-volatile glycosidic aroma precursors. The effects of glucosidases from various origins on the aroma profiles of diverse foods are considered. The research gaps and the future perspectives are examined.
In my opinion, this review is very interesting, valuable, and well-written. However, the manuscript needs some minor changes. Specific comments are as follows:
Line 506: Please change Gewurztraminer to Gewürztraminer
Cis-3-hexenol is written as (Z)-3-hexenol in line 136, please uniform
Lines 146-148: please clarify the meaning of “instant” oolong tea. Does this term refer to the time/temperature of infusion?
Author Response
Thanks for your positive comments.
Line 506: Please change Gewurztraminer to Gewürztraminer
We have done the changes.
Cis-3-hexenol is written as (Z)-3-hexenol in line 136, please uniform
We have done the changes.
Lines 146-148: please clarify the meaning of “instant” oolong tea. Does this term refer to the time/temperature of infusion?
The sentence was rewritten as follows: “Infusion of spray-dried instant Oolong tea led to a notable alteration in the tea's scent profile via glycosides hydrolysis, resulting in an intensified expression of floral, fruity, and grassy notes.”

Reviewer 4 Report
Comments and Suggestions for Authors
The authors propose to summarize in this bibliographic revision (food-2756117) named “Unlocking flavor potential: The role of microbial β-glucosidases in enhancing flavor perception during food processing and in the mouth” the limits and capacities of the beta-glucoside enzymes in the food industry. This specific bibliographic review was written in a manner that does not conform to the standard of scientific review documents because of its lack of focus. The title is one of the examples that the authors must take into consideration; it is redundant and too long; a title like “Unlocking flavor potential using microbial beta-glucosidases in food processing” is good enough. As well, the introduction is too long for a review proposal, the authors must mention the limits and the other reviews close to the proposed manuscript. In other ways, many concepts will be mentioned unnecessarily many times. In the manuscript, a reasonable number of references are cited as evidence of substantive background. The results in general are presented and explained in an unclear and confusing manner, the authors did not show any transversal analysis between the referenced data. The discussion sections proposed by the authors are not as challenging as is expected in this kind of manuscript.
Comments on the Quality of English LanguageThe authors propose to summarize in this bibliographic revision (food-2756117) named “Unlocking flavor potential: The role of microbial β-glucosidases in enhancing flavor perception during food processing and in the mouth” the limits and capacities of the beta-glucoside enzymes in the food industry. This specific bibliographic review was written in a manner that does not conform to the standard of scientific review documents because of its lack of focus. The title is one of the examples that the authors must take into consideration; it is redundant and too long; a title like “Unlocking flavor potential using microbial beta-glucosidases in food processing” is good enough. As well, the introduction is too long for a review proposal, the authors must mention the limits and the other reviews close to the proposed manuscript. In other ways, many concepts will be mentioned unnecessarily many times. In the manuscript, a reasonable number of references are cited as evidence of substantive background. The results in general are presented and explained in an unclear and confusing manner, the authors did not show any transversal analysis between the referenced data. The discussion sections proposed by the authors are not as challenging as is expected in this kind of manuscript.
Author Response
Thanks for your comment, the discussion has been adapted as suggested.

Round 2
Reviewer 4 Report
Comments and Suggestions for Authors
This reviewer does not have further comments.